# BILINEAR VALUE NETWORKS

**Zhang-Wei Hong,**[*] **Ge Yang**[*†] **& Pulkit Agrawal**[†‡]
Improbable AI Lab
NSF AI Institute for AI and Fundamental Interactions (IAIFI)[†], MIT-IBM Watson AI Lab[‡]
Massachusetts Institute Technology

## ABSTRACT

The dominant framework for off-policy multi-goal reinforcement learning involves estimating goal conditioned Q-value function. When learning to achieve multiple goals, data efficiency is intimately connected with generalization of the Q-function to new goals. The de-facto paradigm is to approximate $Q(s, a, g)$ using monolithic neural networks. To improve generalization of the Q-function, we propose a bilinear decomposition that represents the Q-value via a low-rank approximation in the form of a dot product between two vector fields. The first vector field, $f(s, a)$, captures the environment's local dynamics at the state $s$; whereas the second component, $\phi(s, g)$, captures the global relationship between the current state and the goal. We show that our bilinear decomposition scheme substantially improves data efficiency, and has superior transfer to out-of-distribution goals compared to prior methods. Empirical evidence is provided on the simulated Fetch robot task-suite, and dexterous manipulation with a Shadow hand.

## 1 INTRODUCTION

Learning a Q-value function (Watkins & Dayan, 1992) is among the core problems in reinforcement learning (RL) (Sutton & Barto, 2018). It is often desired to learn a goal-conditioned Q-value function (Kaelbling, 1993) that estimates rewards for multiple goals. In multi-goal RL, the Q-function takes the form of $Q(s, a, g)$ which indicates the utility of taking action $a$ at state $s$ for approaching the goal $g$. To enable the agent to work across a wide range of goals, it is usual to train $Q(s, a, g)$ for a large set of goals. Such training requires substantially more data than learning a Q-value function for a single goal (Schaul et al., 2015).

The data efficiency of learning $Q(s, a, g)$ has a close relationship with the generalization of the Q-function to new goals. To understand why, consider the training process where goals are sampled from a pre-defined training distribution. The target of $Q$-learning is to reduce the error on the $Q$-function for all goals in the training set. At an intermediate point in the training process, the agent would have encountered a subset of training goals. Suppose on these goals the error in approximating the $Q$-function is low. If the learned $Q$-function generalizes to new goals sampled in subsequent training, less data would be required to correct the small errors in Q-function for these goals. Conversely, poor generalization would necessitate collecting a large amount of data.

Currently popular methods (Schaul et al., 2015; Andrychowicz et al., 2017) represent the $Q(s, a, g)$ with a monolithic neural network, shown in Figure 1a, whose ability to generalize to new goals is solely derived from the neural network architecture. Because such models jointly process the tuple $(s, a, g)$, the Q-function fails to generalize on new combinations of previously seen individual components. E.g, given training tuples, $(s_A, a_A, g_A)$ and $(s_A, a_B, g_B)$, the monolithic network might fail to generalize to a new data $(s_A, a_A, g_B)$ despite that $s_A$, $a_A$, and $g_B$ have been seen at least once during training.

We proposed **bilinear value networks (BVN)** to improve such generalization by introducing an *inductive bias* in the neural network architecture that *disentangles* the *local dynamics* from the question of *how to reach the goal*. Concretely, we propose to represent the goal-conditioned Q-value

---

[*] Equal contribution, order determined randomly. Authors are also with Computer Science and Artificial Laboratory (CSAIL), and affiliated with the Laboratory for Information and Decision Systems (LIDS). Correspondence to {zwhong,geyang}@csail.mit.edu

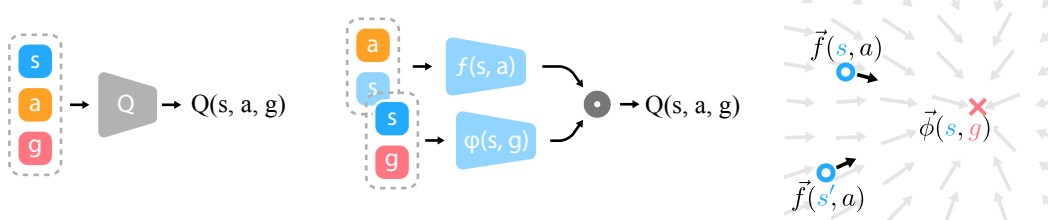

(a) A Monolithic value network    (b) Bilinear value network    (c) Bilinear decomposition

Figure 1: Illustration of why Bilinear Value Networks (BVNs) improve generalization to new goals. (a) The traditional monolithic goal-conditioned value function jointly models the inputs $(s, a, g)$. (b) Bilinear Value Networks (BVN) introduce an inductive bias in the Q-function computation by disentangling the effect of actions at the current state from how to reach the goal. This is realized by representing $Q(s, a, g) = \phi(s, g)^T f(s, a)$, where the two vector fields represent "*what happens next*" $f(s, a)$ and "*where to go next*" $\phi(s, g)$. (c) A 2D navigation toy example, where state $s$ and goal $g$ are both $(x, y)$ positions. The task of this toy example is to reach the goal position $g$. Gray arrows indicate the vector field $\phi(s, g)$ evaluated at all states. The black arrow indicates the best alignment with $f(s, a)$.

function as interaction between two concerns, $\phi(s, g)$ ("*where to go*") and $f(s, a)$ ("*what happens next*"), where each concern can be represented as a neural network. The architecture of BVN is illustrated in Figure 1b. $\phi(s, g)$ models how to change the state to get closer to the goal, while $f(s, a)$ captures where the agent can move from the current state. The Q-function is computed as the alignment between these concerns: $\phi(s, g)^T f(s, a)$. As the $\phi$-network only requires state-goal $(s, g)$ and $f$-network simply needs state-action $(s, a)$, the learned Q-value function can generalize to the new data $(s_A, a_A, g_B)$ since both $(s_A, a_A)$ and $(s_A, g_B)$ are the seen data for $f$ and $\phi$, respectively. Such decomposition is possible because the knowledge of local dynamics ($f$) is independent of where to go ($\phi$) and thus $f(s, a)$ be re-used across goals.

Another intuition about the proposed BVN decomposition is: BVN would predict a high Q-value when the vector representation of the desired future state, $\phi(s, g)$, aligns with the vector representation of the future state, $f(s, a)$, after executing action $a$. We illustrate this intuition with help of a toy example shown in Figure 1c. Here the agent's state $s$ and the goal $g$ are 2D positions, and the task is to get close to $g$ shown as a pink cross. As an analogy to "*what happens next*", the black arrow originating at state $s$ (shown as the blue circle) denotes the succeeding state after taking an action at state $s$, which is goal-agnostic. Independent of action, the grey arrows can be viewed as the direction to the goal from the current state $s$ (i.e., $\phi(s, g)$ or "*where to go*"). While this illustration was in 2D, we find that the proposed bilinear value decomposition substantially improves data efficiency (Section 5.1) and generalization to new goals (Section 5.2) in complex and standard simulated benchmark tasks: (i) dexterous manipulation with a twenty degrees-of-freedom (DoF) shadow hand and (ii) object pick-place tasks using the Fetch manipulator.

## 2 PRELIMINARIES

We consider the multi-goal reinforcement learning problem (Kaelbling, 1993; Plappert et al., 2018), formulated as a goal-conditioned Markov Decision Process (MDP). We assume infinite horizon with the discount factor $\gamma$, and use $\mathcal{S}, \mathcal{A}, \mathcal{G}$ for the state, action, and goal spaces. At the beginning of each episode, an initial state $s_0$ and a goal $g$ is sampled according to the distribution $\rho_{\mathcal{S}}$ and $\rho_{\mathcal{G}}$. The goal $g$ is fixed throughout the episode. At every time step $t$, the agent takes an action according to $a_t \sim \pi(s_t, g)$, and receives the reward $r_{t,g} = R(s_t, a_t, g)$ while transitioning to the next state $s_{t+1}$. The objective of multi-goal RL is to produce a policy $\pi^*$ that maximizes the expected return $J$

$$\pi^* = \arg\max_{\pi} J(\pi) \text{ where } J(\pi) = \mathbb{E}\left[\sum_{\tau=t}^{\infty} \gamma^{\tau-t} r_{\tau,g} | s_t = s\right], \tag{1}$$

where the reward $r_{\tau,g}$ is often structured as the follows:

$$R(s_\tau, a_\tau, g) := \begin{cases} 0, & \text{goal reached} \\ -1, & \text{otherwise.} \end{cases} \quad \forall \tau \in \mathbb{N} \tag{2}$$

Note that our proposed parameterization scheme, BVN, does not rely on this reward structure in general. All of the baselines use a discount factor $\gamma = 0.98$.

## 2.1 OFF-POLICY RL WITH ACTOR-CRITIC METHODS

It is common to use actor-critic algorithms such as *deep deterministic policy gradient* (DDPG) (Silver et al., 2014), *twin-delayed DDPG* (TD3) (Fujimoto et al., 2018) and *soft actor-critic* (SAC) (Haarnoja et al., 2018), to solve multi-goal continuous state-action space tasks considered this paper. In DDPG, the critic $Q$ is learned by minimizing the temporal-difference (TD) loss:

$$\mathcal{L}(Q) = \mathbb{E}_{(s_t, a_t) \sim \text{Uniform}(\mathcal{D})}(r_{t,g} + \gamma Q(s_{t+1}, \pi(s_{t+1}, g), g) - Q(s_t, a_t, g))^2, \tag{3}$$

where $\mathcal{D}$ is the the replay buffer (Mnih et al., 2015) containing state transition tuples $(s_t, a_t, r_t, s_{t+1}, g)$. The actor $\pi$ is trained to maximize the score from the critic, using the deterministic policy gradient:

$$\nabla_\pi J(\pi) = \mathbb{E}[\nabla_a Q(s, a, g)|_{a=\pi(s,g)}]. \tag{4}$$

In multi-goal RL settings, we refer to the goal-conditioned critic as the *universal value function approximator (UVFA)* (Schaul et al., 2015) because $Q(s, a, g)$ approximates the expected utility for all goals $\in \mathcal{G}$. Because the actor relies on the critic for supervision, the ability for the critic to quickly learn and adapt to unseen goals is a key contributing factor to the overall sample efficiency of the learning procedure. A critic that provides better value estimates using less sample and optimization gradient steps, will lead to faster policy improvement via the gradient in Equation 4. For data efficiency it is common to use off-policy actor-critic methods with hindsight experience replay (HER) (Andrychowicz et al., 2017). It is known that TD3 and SAC outperform DDPG on single-task continuous control problems. However, in the multi-goal setting when used along with HER, we found DDPG to outperform both TD3 and SAC on the benchmark tasks we considered (see Section B). We therefore use DDPG for majority of our experiments and report SAC/TD3 results in the appendix.

## 3 BILINEAR VALUE NETWORKS

Bilinear value network (BVN) factorizes the value function approximator into a bilinear form from two vector representations into a scalar:

$$Q(s, a, g) = f(s, a)^\top \phi(s, g), \tag{5}$$

where $f : \mathcal{S} \times \mathcal{A} \mapsto \mathbb{R}^d$ and $\phi : \mathcal{S} \times \mathcal{G} \mapsto \mathbb{R}^d$. The first component $f(s, a)$ is a *local* field that only concerns the immediate action at the current state $s$. The second component is more *global* in the sense that $\phi(s, g)$ captures long-horizon relationship between the current state and the goal.

Closely related to our work is the low-rank bilinear decomposition, $Q(s, a, g) = f(s, a)^\top \phi(g)$, proposed in Schaul et al. (2015). Though Schaul et al. (2015) showed this decomposition attains better data efficiency in simple environments (e.g., gridworlds), this decomposition is not expressive enough to model value functions in complex environments, which we will show

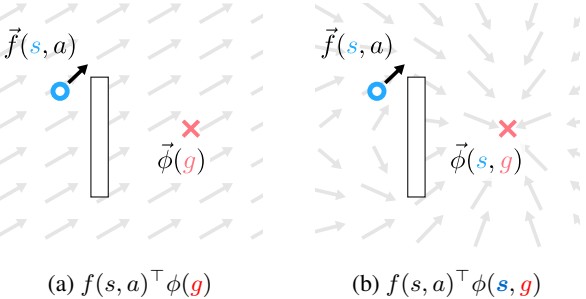

(a) $f(s,a)^\top \phi(g)$      (b) $f(s,a)^\top \phi(\boldsymbol{s},g)$

Figure 2: Comparison between two value decomposition schemes. The task, states, and goals are the same as Figure 1c. The black square indicates obstacles that the agent cannot pass through. Gray arrows indicate the vector field $\phi$ evaluated at a particular goal $g$, for all states $s \in \mathcal{S}$. (a) As in prior work Schaul et al. (2015), the vector field $\phi(g)$ is a constant and does not depend on $s$. (b) BVN parameterizes $\phi$ as a function of both $s$ and $g$ making it more expressive, but still maintaining the benefit of disentanglement from $f(s, a)$.

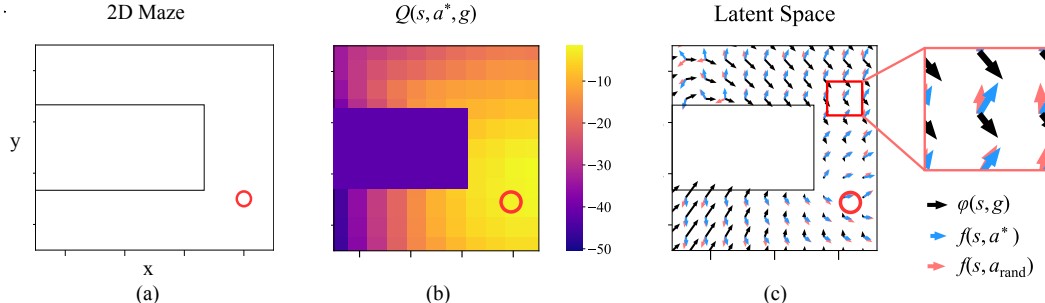

Figure 3: **(a)** The U-maze environment where the states $s$ and goals $g$ are $(x, y)$ on the plane. The task is to reach the goal position (red circle). **(b)** The learned Q-value predictions $Q(s, a^*, g)$ with the best action $a^*$ over states $s$ in the 2-dimensional state space for a goal $g$ indicated by the red circle. Overall, values decrease as the agent gets further and further away from the goal. **(c)** The latent vectors produced by the agent, projected to a 2D vector space. There is an overall trend for $\phi$ to decrease in magnitude as one gets closer to the goal. **(c Inset)** Comparing the alignment of vector representation of a random (suboptimal) action produced by the $f$ component, $f(s, a_{rand})$ (pink) against the vector corresponding to the optimal action from the policy $\pi$, $f(s, a^*)$ (blue) from the $phi$ vector (black). The optimal alignment corresponds to an angle of $90°$ because the maximum Q-value in this environment is 0 and not 1. We find that the optimal action consistently produces a better alignment in comparison to a sub-optimal action.

in Section 5.2. Consequently, the majority of multi-goal RL algorithms use a monolithic network shown in Figure 1a to represent the value function.

To motivate the design of our factorization, we present a vector analysis perspective on previous value decomposition Schaul et al. (2015) and our method in Figure 2, where each point on the 2D plane is a state $s$. In Figure 2a, we illustrate the vector field of low-rank bilinear UVFA (Schaul et al., 2015), where $f : \mathcal{S} \times \mathcal{A} \mapsto \mathbb{R}^d$ and $\phi : \mathcal{G} \mapsto \mathbb{R}^d$. It can be seen that $\phi(g)$ is a constant across states since $\phi(g)$ does not depend on the state $s$ in Figure 2a. This makes learning more challenging due to restricted expressivity. The expressivity of low-rank bilinear UVFA is restricted since the such a decomposition can only model linear relationships between states and goals. Consequently, the burden of modeling different Q-values for different states falls completely on $f$. In contrast, in Figure 2b, the field $\phi(s, g)$ produced by BVN is also a function of the state $s$. This makes $\phi$ in BVN strictly more expressive than low-rank bilinear UVFA that do not depend on $s$.

Bilinear value networks are a drop-in replacement for the monolithic neural networks for approximating universal value function approximator. Our implementation is not specific to a method and be used to learn value function using a variety of methods such as DDPG, SAC, TD3 or even methods that we do not make use of temporal differenceing.

## 3.1 A TOY EXAMPLE: VISUALIZING $f(s, a)$ AND $\phi(s, g)$

To examine whether the learned bilinear value decomposition reflect our intuition mentioned in Section 1, we trained a deep deterministic policy gradient (DDPG) (Lillicrap et al., 2016) with hindsight experience replay (HER) (Andrychowicz et al., 2017) agent using BVN in a 2D maze shown in Figure 3 (a), where $s \in \mathbb{R}^2$, $a \in \mathbb{R}^2$, $g \in \mathbb{R}^2$, and the reward is the negative euclidean distance between states and goals. The output dimensions of $f$ and $\phi$ are both 16.

We computed the vectors $z_f^* = f(s, a^*)$ and $z_\phi = \phi(s, g)$ densely on a grid of states $(x, y)$ with respect to the fixed goal $g$, where $a^* = \pi(s, g)$ is the optimal action predicted by the learned policy. For visualization, we used principal component analysis (PCA) to project $z_\phi$ to a 2-dimensional vector and is plotted as black arrows in Figure 3(c). We cannot use PCA to plot $z_f^*$ on the 2D plane along with $z_{phi}$ because $f(s, a^*)$ and $\phi(s, g)$ live in two different vector spaces. Instead, we plot $f(s, a^*)$ as a rotation from $z_\phi$ denoted as $\angle(f(s, a^*), \phi(s, g))$. Specifically, we first plot the 2D projection of $z_{phi} = \phi(s, g)$ as a 2D vector and then plot $z_f^* = f(s, a^*)$ by rotating the plotted $z_\phi$ by angle of $\angle(f(s, a^*), \phi(s, g))$.

Further, Figure 3(c) shows that the length of $z_{phi}$ correlates to the distance between the state ($s$) and the goal ($g$), providing further evidence in support of our hypothesis that $\phi(s, g)$ captures proximity to the goal. If $f(s, a)$ indeed captures the dynamics, then $z_f^*$ (i.e., vector representation of future state after executing the optimal action $a^*$) should be closely aligned to $z_{phi}$ to produce the maximal Q-value. To test if this is the case, we also plot in pink $z_f^{rand}$ denoting the vector representation of future state after executing a random action $a^{rand}$. We find that in comparison to $f(s, a^{rand})$, $f(s, a^*)$ better aligns with $\phi(s, g)$ validating our hypothesis. Note that the optimal alignment corresponds to an angle of $90°$ because the maximum Q-value in this environment is 0 and not 1. As a result, $f$ for the optimal action will be orthogonal to $\phi$.

## 4   EXPERIMENTAL SETUP

**Tasks**   We evaluate our method in all the multi-goal RL benchmarks in  Plappert et al. (2018), *Fetch* robot and *Shadow hand* manipulation tasks, as shown in Figure 10. For *Fetch* robot domain, the robot is required to move the object to a target position. In *Shadow hand* domain, the agent controls the joints of the hand's fingers and manipulates the objects in hand to a target pose. For all tasks, we set discount factor $\gamma = 0.98$ based on Plappert et al. (2018). The details, including the reward functions, of these tasks can be found in the Section A.2.

**Baselines**   We compare with two baselines, *UVFA (Monlithic)* (Andrychowicz et al., 2017; Schaul et al., 2015) and *UVFA (Bilinear)* proposed in Schaul et al. (2015). *UVFA (Monlithic)* is the most common multi-goal value function architecture that parametrizes the Q-function by a monolithic neural network. *UVFA (Bilinear)* is a bi-linear value function that decomposes the Q-function as $Q(s, a, g) = f(s, a)^\top \phi(g)$, where $f$ and $\phi$ denote two neural networks respectively. Our method uses a different bilinear decomposition and we refer to it as *BVN (Ours)*.

**Implementation**   The baselines and our method are trained by deep deterministic policy gradient (DDPG) (Lillicrap et al., 2016) with hindsight experience replay (HER) (Andrychowicz et al., 2017). We use DDPG with HER as the base algorithm for our method and the baselines because it outperforms other recent Q-learning algorithms (Haarnoja et al., 2018; Fujimoto et al., 2018) on the task of learning a goal-conditioned Q-function. Comparison of DDPG against SAC (Haarnoja et al., 2018) and TD3 (Fujimoto et al., 2018) is provided in Section B. For all baselines, the Q-function is implemented as a three layer neural network with 256 neurons per layer. In our method, $f$ and $\phi$ are both implemented as a 3-layer neural network with 176 neurons to match the total number of parameters of the baselines' Q-function. For *BVN (Ours)* and *UVFA (Bilinear)*, the output dimensions $f$, $\phi(s, g)$, and $\phi(g)$ are set as 16. The rest of hyperparameters for training are left in Section A.3.

**Evaluation metric**   We ran each experiment with 5 different random seeds and report the mean (solid or dashed line) and 95%-confidence interval (shaded region) using bootstrapping method (Di-Ciccio & Efron, 1996). Each training plot reports the average success rate over 15 testing rollouts as a function of training epochs, where the goal is sampled in the beginning of each testing rollout. Each epoch consists of a fixed amount of data (see Section A.3 for details).

## 5   RESULTS

Our experiments answer whether BVN improves sample efficiency and generalizes better to unseen goals. In Section 5.3, we provide ablation studies probing the importance of several design decisions.

### 5.1   SAMPLE EFFICIENCY AND ASYMPTOTIC PERFORMANCE

Figure 4 compares the sample efficiency of our method, *BVN (Ours)*, against baselines on the *Fetch* tasks. The results show that our method has higher data efficiency in four out of eight tasks and matches the baselines in other tasks. Our method also attains higher asymptotic performance in *Bin-place*. Figure 5 shows that on the *Shadow hand* task suite, BVN improves the sample efficiency in six out of the eight tasks and is on-par with the baselines for the other two tasks. These results suggest that BVN consistently improves sample efficiency of learning goal-conditioned policies.

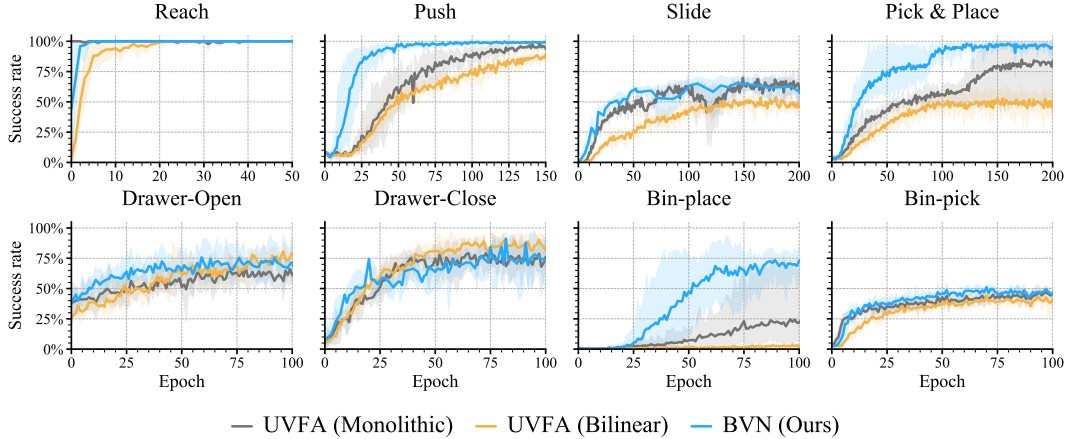

Figure 4: Learning curves on the *Fetch* gym environments, and the *Fetch-extension* (see Section A.4) task suit. BVN attained higher success rates using less data than the baselines in 5 out 8 domains. Also, BVN achieved higher asymptotic performance in *Bin-place*.

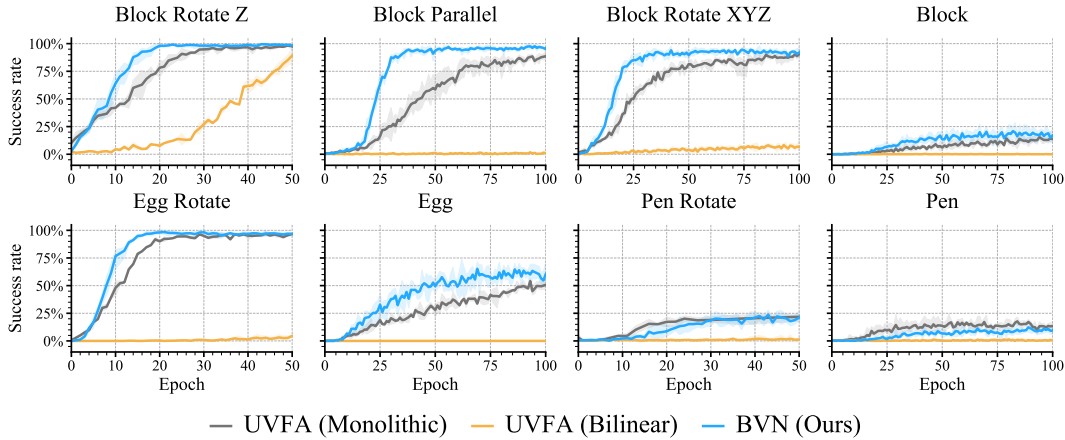

Figure 5: Learning curves on the *Shadow hand* dexterous manipulation suite. BVN learns faster than the baselines on six out of eight domains.

## 5.2 GENERALIZATION TO UNSEEN GOALS

Our central hypothesis is that BVN increases data efficiency by generalizing better to new goals. To test this, we split the training and evaluation goals in two ways in the *Fetch* robot domain: (1) left-to-right and (2) near-to-far. In near-to-far scenario, all goals within a pre-specified euclidean distance are used for training (*near*) and goals sampled outside the threshold distance are used for testing (i.e., *far*). The train-test split of goals is illustrated in Figure 6b by colors green and pink respectively. For left-to-right scenario, we split the goal space into the left and right half-spaces (see Figure 6a). During training, we disable HER to prevent the Q-function from receiving goals in the testing split due to relabeling. We train the baselines and our method until they achieve similar success rates (see Section A.3 for details). For testing, we finetune the learned Q-function and the policy using the weights from the best model over 3 random seeds from training time. We finetune the Q-function and the policy using HER.

The fine-tuning performance of the baselines and BVN is reported in Figure 6c. We find that BVN (shown in blue) quickly adapts to new goals and outperforms the monolithic UVFA (shown in grey). This result suggests that BVN generalizes better than *UVFA (Monlithic)* to new goals.

Recall that in the BVN formulation, $f(s, a)$ does not encode any goal specific information. We therefore hypothesized that only finetuning $\phi(s, g)$ should be sufficient for generalizing to new goals. We call this version of finetuning as *BVN (freeze f)*. Results in Figure 6c support our hypothesis

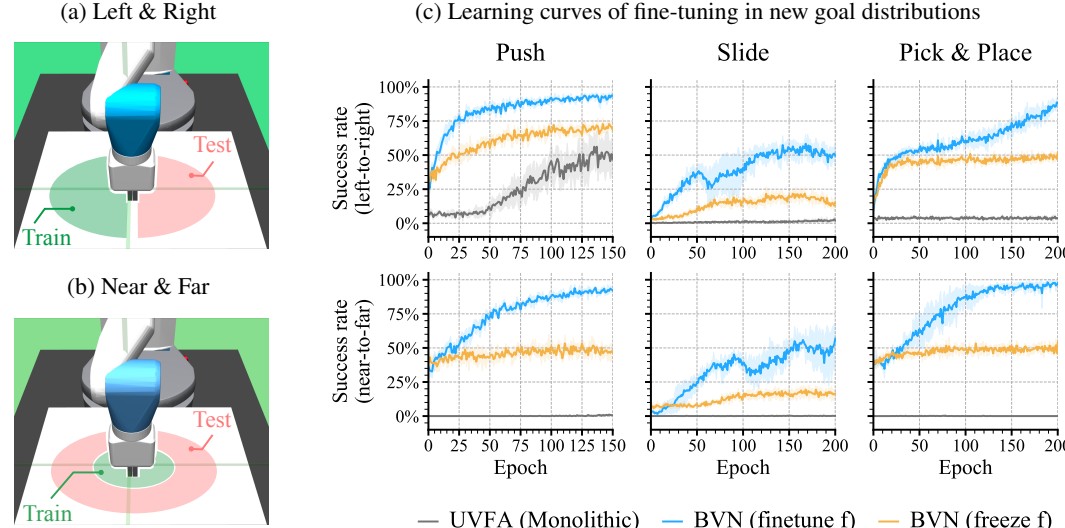

Figure 6: **(a & b)** Examples showing (a) how we split the goal distribution into left and right and (b) according to the radius to the center (near versus far). We pretrain on the green regions, then finetune on the red regions. The state distribution remain identical to the vanilla *Fetch* environment. **(c)** Fine-tuning curves on unseen goals. BVN achieves better performance than the *UVFA (Monlithic)* on all tasks in both left-to-right and near-to-far adaptation scenarios. Freezing $f$ causes BVN performance to plateau, indicating that both $\phi$ and $f$ are needed for approximating the multi-goal value function.

as *BVN (freeze f)* (orange) indeed outperforms *UVFA (Monlithic)*. However, there is a significant performance gap between *BVN (finetune f)* and *BVN (freeze f)*. This result indicates that while BVN provides an inductive bias to disentangle the local dynamics, $f(s, a)$, from the goal-specific component, $\phi(s, g)$, such disentanglement is not fully learned. This could be due to several reasons such as learning on limited data or because the disentanglement between $f$ and $\phi$ is under-specified upto a scalar factor (i.e., $\lambda f$ and $\frac{1}{\lambda}\phi$ yield the same result for a scalar $\lambda$), etc. Nonetheless, the results convincingly demonstrate that the proposed bilinear decomposition increased data efficiency and improves generalization. The topic of achieving perfect disentanglement is outside the scope of current work and a worthwhile direction for future research.

## 5.3 Ablation studies

**How does latent dimension affect our method?** Though the dimensions of the latent space (i.e., output dimension of $f$ and $\phi$) are set as 16 across all the experiments above, we investigate the effect of the dimensionality of the latent space on the agent performance. Figure 7 shows the learning curve for BVN with 3, 4, 8 and 16 dimensional latent spaces. The performance consistently improves as latent dimensions increase, and even a BVN with just a 3-dimensional latent space still outperforms the monolithic UVFA baseline. Hence bilinear value decomposition produces consistent gain with respect to the monolithic UVFA baseline regardless of the latent dimensions, and benefits performance-wise from a larger latent space.

**Dot product "·" vs $\ell^2$ Metric $\|\cdot\|_2$** As it is unclear if the dot-product is the best choice to model the interaction between two vector fields $f$ and $\phi$, we compare two alternative ways to decompose the value function using the same variable grouping as the BVN. In the first variant, we replace the dot-product with an $\ell^2$ metric $\|\cdot\|_2$, and parameterize the Q function via $Q(s, a, g) = -\|f(s, a) - \vec{\phi}(s, g)\|_2$. The negative sign is added because multi-goal RL tasks typically assume the reward is within the range $[-1, 0]$. We set the latent dimension to be *three* as it is the best one found through hyperparameter search (see Section A.4). In the second variant we linearly combine the two latent vectors: $Q(s, a, g) = w^i f_i(s, a) + v^i \phi_i(s, g)$, where $x^i y_i$ implies a sum over the $i$ indices implicitly. This is the Einstein notation, or "einsum". The results shown in Figure 8 shows that both the bilinear

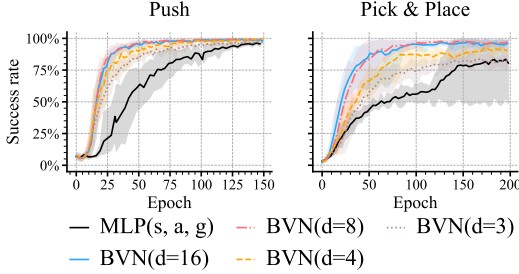
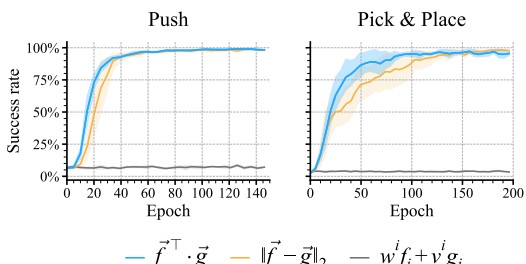

Figure 7: Increasing the latent dimension of $f$ and $\phi$ improves our method's performance. Despite that, the result shows that a low dimensional latent space ($d = 3$) is sufficient to enable our method to outperform the baseline *UVFA (Monlithic)*.

Figure 8: Replacing the dot-product with an $\ell^2$ norm does not affect the final performance or the sample complexity. The initial performance is affected marginally. Whereas the linear combination is too restrictive.

and the metric decomposition attain good performance. The $\ell^2$ metric decomposition is marginally slower during the initial learning phase. The linear combination works poorly, likely due to the restricted expressivity.

**Alternate Bi-Linear Decomposition**   Lastly, we investigate if there are other viable ways to decompose the Q-function based on grouping the inputs $s, a, g$. An alternative hypothesis is that any grouping of the inputs arguments (i.e., $s, a, g$) could improve the sample efficiency. To test if this is the case, we compare our bilinear value decomposition scheme against two alternatives, $f(s, a)^\top \phi(a, g)$ and $f(s, a, g)^\top \phi(g)$, as well as the scheme $f(s, a)^\top \phi(g)$ shown in Section 5.1. The results in Figure 9 show that neither matches the performance of the bilinear value network, which suggests correct grouping does play a crucial role. It is important to process $s$ and $g$ together and duplicate the state input $s$ on both branches.

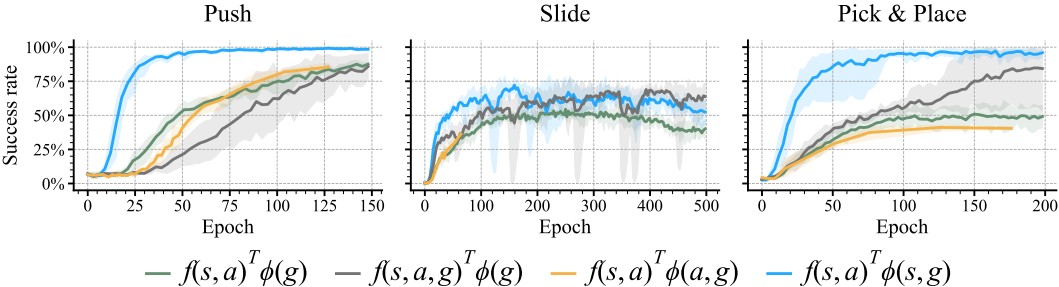

Figure 9: Comparison between alternative factorizations show that our bilinear value decomposition scheme, $f(s, a)^\top \phi(s, g)$, outperforms these alternatives. This suggests that this particular input grouping is crucial.

## 6   RELATED WORKS

A large number of prior works have explored ways to decompose value function to improve sample efficiency of value function approximation. Classic works (Sutton & Barto, 2018; Littman et al., 2001; Singh et al., 2003) approximate the value function in a set of linear basis. Successor features (Dayan, 1993; Kulkarni et al., 2016; Borsa et al., 2019) factorize the value function as the dot product between expectation of state-action feature occupancy and a vector of reward coefficients to the features.r Laplacian reinforcement learning (Mahadevan & Maggioni, 2006; 2007) represents the value function using a set of Fourier basis. The above previous works investigate value function decomposition in single task learning. Instead, bilinear value networks is focused on decomposing a multi-goal value function by disentangling the representations for goals and actions. Also, bilinear

value networks use end-to-end training to learn the decomposition instead of separating the basis learning and the value approximation. To summarize prior works and their choice of parameterization, we compile key publications in Table 1, ordered by the year of publication, in the appendix.

## 7 CONCLUSION AND DISCUSSION

Our empirical evaluation shows significant sample efficiency gain when bilinear value networks is used with multi-goal RL tasks. Moreover, bilinear value networks does not introduce extra hyperparamters for training, except for the output dimensions of $f$ and $\phi$. Our ablations demonstrate that the benefits of BVN result from faster adaptation of the value function to new goals. An exciting direction for future research is to facilitate transfer across different robots' morphology and goal spaces. One could keep the state and goal spaces fixed and train a single $\phi(s, g)$ that is shared between the different morphologies. The difference between morphologies could be accommodated by different $f$ (e.g., different manipulators for the same task). Also, one can pre-train a shared $f$ for the same robot and train $\phi$ for different tasks (i.e., goal spaces). In addition, we further discuss the additional implications of our experimental results in Section C of appendix.

### ACKNOWLEDGMENTS

We thank members of Improbable AI Lab for the helpful discussion and feedback. We are grateful to MIT Supercloud and the Lincoln Laboratory Supercomputing Center for providing HPC resources. The research in this paper was supported by the MIT-IBM Watson AI Lab, in part by the Army Research Office and was accomplished under Grant Number W911NF-21-1-0328 and by the National Science Foundation AI Institute for Artificial Intelligence and Fundamental Interactions ( https://iaifi.org/) under Cooperative Agreement PHY-2019786. The views and conclusions contained in this document are those of the authors and should not be interpreted as representing the official policies, either expressed or implied, of the Army Research Office or the United States Air Force or the U.S. Government. The U.S. Government is authorized to reproduce and distribute reprints for Government purposes notwithstanding any copyright notation herein.

## 8 REPRODUCIBILITY STATEMENT

We provide detailed instructions for reproducing the results in this paper in the Appendix. Please refer to Section Section A.4.

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

Table 1: Comprehensive list of linear value decomposition schemes from the literature.

| Reference | Parameterization |
|---|---|
| Linear basis (Sutton & Barto; Littman et al.; Singh et al.) | $Q(s,a) = \sum c_i Q^i(s,a)$ |
| Proto-Value Function (Mahadevan & Maggioni, 2006; 2007) | $V(s) = \sum \lambda_i V^i(s)$ |
| General Value Approximator (GVF/Horde, Sutton et al. 2011) | $Q(s,a,\theta) = \theta^\top \phi(s,a)$ |
| Successor Representation (SR, Kulkarni et al. 2016) | $Q(s,a) = M(s,a,s')R(s')$ |
| Monolithic MLP UVFA (Schaul et al.; Andrychowicz et al.) | $\text{MLP}(s,a,g)$ |
| Recurrent UVFA (Mirowski et al., 2017) | $\text{RNN}(z_s, a, z_g), z_s, z_g$ are embeddings |
| Low-rank Bilinear UVFA (Schaul et al., 2015) | $Q(s,a,g) = f(s,a)^\top w(g)$ |
| USFA (Borsa et al., 2019) | $Q(s,a,\vec{z}) = f(s,a,\vec{z})^\top \vec{w}$ |
| Bilinear Value Network (**ours**) | $Q(s,a,g) = f(s,a)^\top \phi(s,g)$ |

## A  APPENDIX

### A.1  DETAILS ON PREVIOUS LINEAR DECOMPOSITION SCHEMES

We present the comparison of parameterizations in Table 1.

### A.2  IMPLEMENTATION DETAILS OF TASKS

All experiments in Section 5.1 happen on the standard object and dexterous manipulation tasks from the gym robotics suite (Plappert et al., 2018).

We briefly summarize the specifications of the tasks in the list below and illustrate these tasks in Figure 10:

Table 2: Environment Specifications.

| Task | Fetch Robot $\mathcal{S}$ | $\mathcal{A}$ | $\mathcal{G}$ | Task | Shadow Hand $\mathcal{S}$ | $\mathcal{A}$ | $\mathcal{G}$ |
|---|---|---|---|---|---|---|---|
| Reach | $\mathbb{R}^{10}$ | $\mathbb{R}^3$ | $\mathbb{R}^3$ | BlockRotateZ | $\mathbb{R}^{61}$ | $\mathbb{R}^{20}$ | $\mathbb{R}^7$ |
| Push | $\mathbb{R}^{25}$ | $\mathbb{R}^3$ | $\mathbb{R}^3$ | BlockRotateParallel | $\mathbb{R}^{61}$ | $\mathbb{R}^{20}$ | $\mathbb{R}^7$ |
| PickAndPlace | $\mathbb{R}^{25}$ | $\mathbb{R}^4$ | $\mathbb{R}^3$ | BlockRotateXYZ | $\mathbb{R}^{61}$ | $\mathbb{R}^{20}$ | $\mathbb{R}^7$ |
| Slide | $\mathbb{R}^{25}$ | $\mathbb{R}^3$ | $\mathbb{R}^3$ | BlockRotate | $\mathbb{R}^{61}$ | $\mathbb{R}^{20}$ | $\mathbb{R}^7$ |
| DrawerOpen | $\mathbb{R}^{25}$ | $\mathbb{R}^3$ | $\mathbb{R}^3$ | Egg | $\mathbb{R}^{61}$ | $\mathbb{R}^{20}$ | $\mathbb{R}^7$ |
| DrawerClose | $\mathbb{R}^{25}$ | $\mathbb{R}^3$ | $\mathbb{R}^3$ | EggRotate | $\mathbb{R}^{61}$ | $\mathbb{R}^{20}$ | $\mathbb{R}^7$ |
| | | | | Pen | $\mathbb{R}^{61}$ | $\mathbb{R}^{20}$ | $\mathbb{R}^7$ |
| | | | | PenRotate | $\mathbb{R}^{61}$ | $\mathbb{R}^{20}$ | $\mathbb{R}^7$ |

In addition to (Plappert et al., 2018), we also use the tasks from Fetch extension[1]. For left/right environments used in Section 5.2, the goal $(x, y, z)$ with $x < 0$ are left goals, and $x \geq 0$ right goals. In near/far environments, we set the distance thresholds as $0.2$ for Slide, $0.2$ for PickAndPlace, $0.1$ for Reach, and $0.15$ for Push. The goals with distance below the threshold are considered as near goals, otherwise far goals.

### A.3  TRAINING HYPERPARAMETERS

**Fetch**  We adapted the hyperparameters in (Plappert et al., 2018) for training in a single desktop. In Plappert et al. (2018), the agent is trained by multiple workers collecting data in parallel, where each worker collects rollouts using the same policy with different random seeds. The performances of the agent trained by our hyperparameters match that in (Plappert et al., 2018).

- num_workers: 2 for Reach, 8 for Push, 16 for Pick & Place, and 20 for Slide.
- Batch size: 1024

---

[1]https://github.com/geyang/gym-fetch

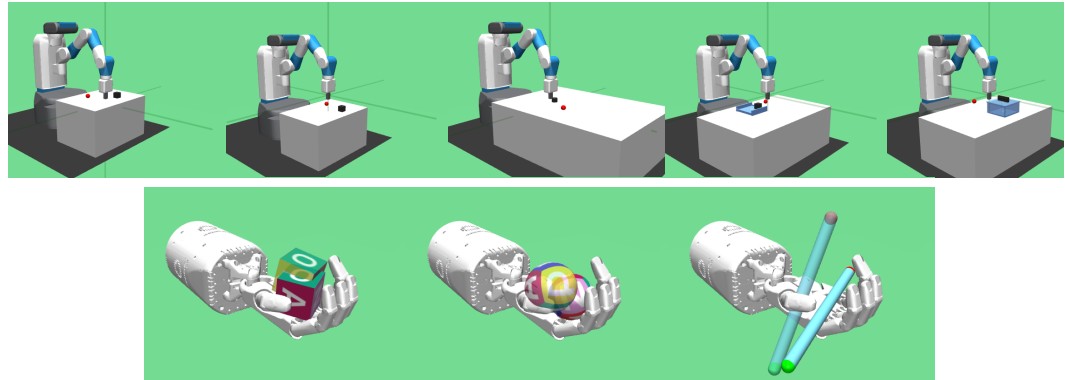

Figure 10: Renderings of the *fetch robot* and *shadow hand* domains considered in this work. Left-to-right: (**top**) Fetch push, pick-and-place, slide, bin-pick, box-open; (**bottom**) Shadow hand block, egg, pen. For Fetch, the goal is to move the gray object to the red dot by controlling the arm. For Shadow hand, the objective is to reorient the object (e.g., block, egg, and pen) to the desired poses rendered by the transparent object.

Table 3: The success rates of each method in the pretrained tasks used in Section 5.2.

|  | Left | | | Near | | |
|---|---|---|---|---|---|---|
|  | Push | Pick & Place | Slide | Push | Pick & Place | Slide |
| UVFA (Monolithic) | 96% | 50% | 81% | 93% | 91% | 96% |
| BVN | 96% | 59% | 48% | 95% | 96% | 90% |

- `Warm up rollouts`: We collected 100 initial rollouts for prefilling the replay buffer.
- `Training frequency`: We train the agent per 2 environment steps. Each iteration of training uses the `num_workers` batches for training.

The rest of hyperparameters remain the same as that used in (Plappert et al., 2018).

**Shadow hand**   We use the hyperparameters used in (Plappert et al., 2018) and reuse the open source code[2].

**Adaptation experiments (Section 5.2)**   We present the success rates after pretraining in Table 3. Note that in the pretraining stage of each method and task, we trained the agents with 5 different random seeds and took the best-performing models as the initialization at the fine-tuning time.

A.4   IMPLEMENTATION DETAILS

**Q-function architecture**   The monolithic network baselines in this paper represents the Q-function as a multi-layer perceptron (MLP) network that takes in $s$, $a$, and $g$ concatenated together as a single vector. The bilinear UVFA (Schaul et al., 2015) uses an MLP $f$ that takes the concatenated $s$ and $a$ as inputs, and an MLP $\phi$ using $g$ as inputs. Our biliniear value network on the other hand, concatenate $[s, a]$ together as the input to $f$, and $[s, g]$ as the inputs of $\phi$.

**Policy architecture**   We use an MLP as the policy network for all the methods.

For our method and its variants presented in Section 5.3 , we use the following nomenclature to denote them:

1. **Dot product** This is our proposed method. We set the outputs' dimension of $f$ and $\phi$ as 16.

---

[2]https://github.com/TianhongDai/hindsight-experience-replay

2. **2-norm** We parametrize Q-function as $Q(s, a, g) = \|f(s, a) - \phi(s, g)\|_2$. We use set the dimension of the outputs from $f$ and $\phi$ as 3. Figure 11 shows that $d = 3$ works the best for 2-norm.

3. **concat** We concatenate $f(s, a)$ and $\phi(s, g)$, feed the concatenated vectors to an MLP layer, and take the outputs of this layer as the Q-value $Q(s, a, g)$. The dimensions of the outputs of $f$ and $\phi$ are 16.

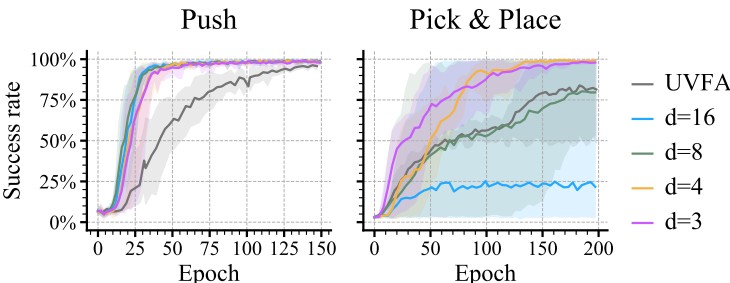

Figure 11: 3-dimensional latent space works the best for 2-norm variant (see Section A.4).

### A.5 TRANSFERRING ACROSS TASKS

In multi-goal RL tasks, we observe that most of tasks are identical in terms of high-level goals, but they have different environment dynamics. For instance, the objective of "Push" and "Slide" are all to move the object to the destination. They only differ in the friction coefficient of the table where the agent cannot slide objects in "Push" because of high friction. Because we decouple the state-action and goals into two different functions, $f(s, a)$ and $\phi(s, g)$, one potential benefit on multi-goal RL is reusing $\phi$ for transferring to tasks with the same (or similar) task objectives (i.e., reward specification). Reusing $\phi(s, g)$ exploits the learned reward-specific state-goal relations. Then we simply need to learn a new $f$ to recover the reward-agnostic relations in the new task. To examine this hypothesis, we select three tasks with the same state and action spaces[3], and measure the performance of transferring amongst these tasks. In transferring an agent from a source task to a target task, we initialize *UVFA (Monlithic)* and *BVN (no reset)* agent's policy and Q-function by the neural networks' weights learned in the source task and fine-tune them in the target task. For *BVN (reset f)*, we reinitialize $f$ and fine-tune $\phi$ when training them in the target task. Figure 12 shows the success rates of transferring from a source task to a target task. For example, "Push to Slide" indicates transferring from "Push" to "Slide". It can be seen that *BVN (reset f)* attains higher success rates at convergence than the baselines and *BVN (no reset)* in 4 out 6 tasks (i.e., "Push to Slide", "Pick & Place to Push", "Push to Pick & Place", and "Pick & Place to Slide").

The limited efficiency gain could result from that $\phi$ is dependent on states $s$ and thus $\phi$ cannot be effectively reused in new tasks. Even though the target task's reward function is the same as that in the source task, the relation of states and goals can be affected by the dynamics. For instance, suppose we have a state where the object is near the goal. In "Pick & Place", because the friction coefficients of the table are high, the $\phi(s, g)$ should direct the agent to lift the object. On the other hand, in "Slide", the $\phi(s, g)$ is expected to guide the agent to push the object to slide.

---

[3]If the two tasks have different state/action spaces, we will not be able to use the weights from a task and resume training in another task.

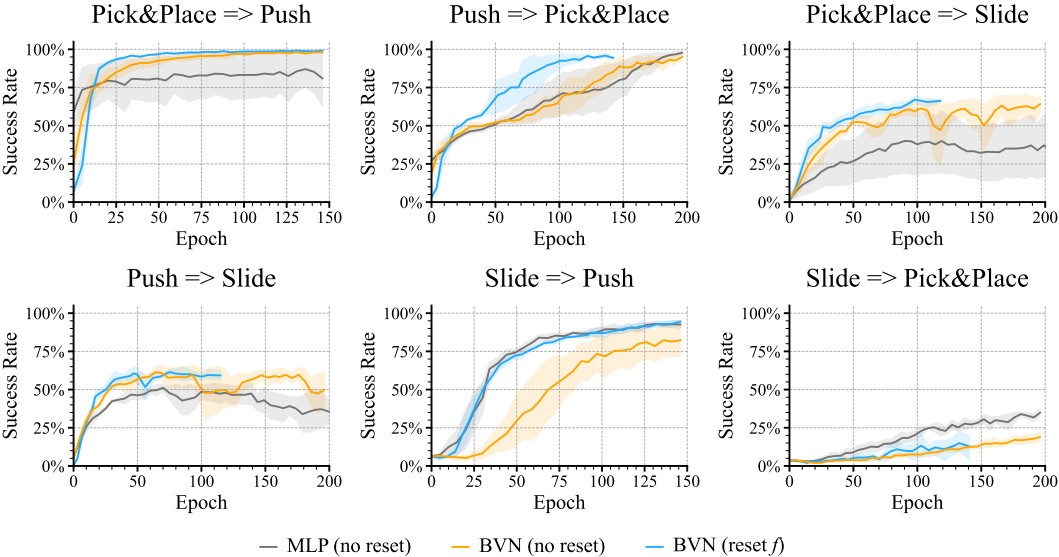

Figure 12: The two-stream architecture of the BVN allows us to improve transfer by resetting $f$. Resetting $f$ and fine-tuning $\phi$ achieve higher success rate than the baselines when transferring between push and pick & place, and to slide.

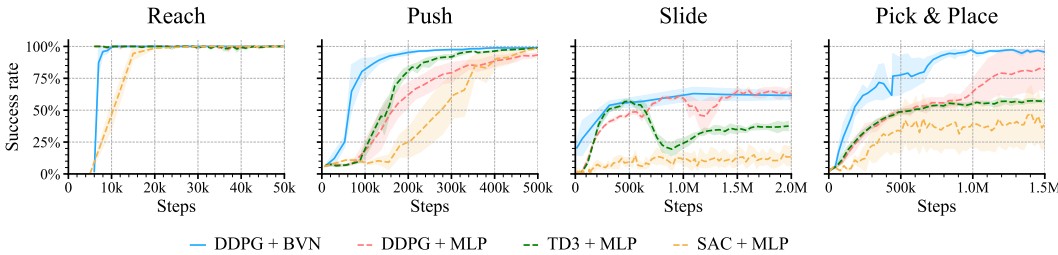

Figure 13: We show that our BVN improves the sample efficiency over the best baseline, *DDPG + MLP*, in 4 commonly used environments in multi-goal RL.

## B  COMPARISON AGAINST SOTA ALGORITHMS

Bilinear value networks achieves State-Of-The-Art (SOTA) performance in common multi-goal reinforcement learning tasks (Plappert et al., 2018). In figure 13, we compare bilinear value networks against two strong baselines: soft actor critic (SAC, see Haarnoja et al. 2018) and twin-delayed deep deterministic policy gradient (TD3, see Fujimoto et al. 2018). We reuse the hyperparameters reported in Fujimoto et al. 2018) and implement TD3 in our codebase. As for SAC, we use the implementation in this codebase[4]. Figure 13 shows the performance of each baseline algorithm and our method (*DDPG + BVN*). We show that our *DDPG + BVN* outperforms the best baseline, and find that *DDPG* works the best among all the *MLP* variants. Our findings align with (Pitis et al., 2020) and the documentation in its released code[5]. Pitis et al. (2020) also found that DDPG works better than TD3 and SAC in multi-goal RL domains.

## C  ADDITIONAL DISCUSSION

**Why does other parameter grouping schemes fail?** In Section 5.3, we showed that $f(s, a, g)^T \phi(g)$ inputs grouping fails to improve sample efficiency. One explanation to this result is that $f(s, a, g)$ cannot prevent negative interference from new goals encountered during training

---

[4]https://github.com/spitis/mrl

since goals $g$ couple with states-actions $s, a$. As a result, the merit of bi-linear value function decomposition vanishes and even causes extra burden on learning because the number of model parameters is larger than *UVFA (Monlithic)*. This preliminary finding shows that to benefit from decomposition, the guideline is to only expose necessary inputs to each component. Redundant inputs in either component could break the decomposition.

**How is 2-norm compared with dot product?**   Section 5.3 demonstrated that using 2-norm to reduce $f(s, a)$ and $\phi(s, g)$ into Q-value matches the performance of our BVN with even lower dimensions of $Z_f$ and $Z_\phi$. However, 2-norm relies on an assumption that the reward function must be non-positive or non-negative since norm is defined as a non-negative value. The $\| f(s, a) - \phi(s, g) \|$ can only fit to either non-positive or non-negative Q-value by manipulating the sign of norm. On the other hand, dot product is more expressive and free from the reward function assumption.

**Can our method facilitate policy transfer?**   Since our method decomposes the value function into a local (i.e., goal-agnostic) and a global (i.e., goal-specific) component, our BVN might facilitate policy transfer by resetting either one of components and fine-tuning another one. As multi-goal tasks considered in (Plappert et al., 2018) have the same objective in nature but different dynamics, one potential advantage of our method is to reuse $\phi(s, g)$ in the new task and re-train $f(s, a)$. The results in Section A.5 showed that the agent transferred by BVN converges to the the performance of training from scratch in the target task slightly faster than the baselines. One reason for the limited policy transferring performance is that $\phi(s, g)$ conditions on states $s$. Consequently, environment dynamics shifts could affect the representations of $\phi$. We, nevertheless, showed that states variables are crucial in $\phi$. One exciting avenue is to disentangle the representations for keeping the rich expressivity as well as desired modularity for transferring across tasks.

