# OpenReview forum: "Bi-linear Value Networks for Multi-goal Reinforcement Learning"
_ICLR.cc/2022/Conference — ICLR 2022 Poster_

### Official Review · Reviewer_QczY · 2021-11-01

**Correctness:** 4
**Technical Novelty And Significance:** 3
**Empirical Novelty And Significance:** 3
**Recommendation:** 8
**Confidence:** 4

**Main Review:**

## Main Review
This paper studies the structure of the function approximators used to learn goal-conditioned policies.

### Strengths
* The paper presents a valid hypothesis that a structure that separates the task-specific information from the action-specific effects should lead to better learning. Their rationalization for this hypothesis is also clear.
* The particular approach this paper takes to effect such a separation is fairly straightforward.
* The difference compared to the bi-linear approach suggested in [1] is also clear.
* The experiments validate one part of the hypothesis by showcasing faster learning with the separated networks on a few of the domains tested on, as well as better transfer to goals in parts of the state space that the agent has not trained on.

### Weaknesses
* The paper presents a clear hypothesis for how they expect the separate networks to learn different aspects of the problem, but then do not construct or find a simpler domain where this hypothesis can be evaluated. Specifically, seeing an experiments that learns a vector field like Fig 2 (b) in actuality would have strengthened the case for the paper quite a bit.
* The idea of separating the learning into what can be accomplished by the agent in a given state and what needs to be done for a given goal could be generalized and analyzed more carefully and deeply than has been done in this paper. As it stands, this paper shows some interesting empirical results with a new network architecture and presents a credible hypothesis for why this architecture might be helping learning, but does not dig deep enough to validate the hypothesis. It also does not attempt to generalize and extract insights beyond what are applicable to the few domains that it evaluates on.
* The paper mentions that the value network is decomposed into the bilinear networks the paper studies. But how are the policy networks handled? Are they keeping their monolithic structure? If so, why? These questions are not answered in the paper effectively.

### Minor issues
* The problem setup seems imprecise. A reader not familiar with reinforcement learning might not be able to parse this section. Specifically the structure of the reward function is not clarified, not to mention the range of discounting is unclear.
* The writing of the paper is a bit imprecise in other places as well. In the experimental section, it refers to [2] as UVFA and [1] as a bilinear network, while the title of paper [1] is "Universal Value Function Approximators". It does so in other parts of the paper as well, where the implication becomes that a UVFA has to be a monolithic network.
* The discussion section seems like an extension of the experimental section. There is a variety of possible generalizations and implications of this work that should be discussed which are ignored in the discussion section for going over a few more experimental details.
* The attribution of [2] should be for the NeurIPS version, rather than the ArXiv one.

### Other points
* The idea of separating what can be done in a state and what needs to be done for a task bears some resemblance to the Q(s, s') learning in [3]. Perhaps it might be useful for discussion.

## References
[1] Universal value function approximators, Schaul et al., ICML 2015
[2] Hindsight Experience Replay, Andrychowicz et al., NeurIPS 2017.
[3] Estimating Q (s, s') with Deep Deterministic Dynamics Gradients, Edwards et al., ICML 2020.

**Summary Of The Paper:**

This paper considers the problem of learning a goal-conditioned policy effectively. Particularly, it looks at the architecture of the value network used to learn this goal-conditioned policy if using a deterministic policy gradient algorithm.
It studies whether separating the goal-conditioned value network into two components, a state-action embedding network and a state-goal embedding network, might lead to faster learning and better generalization of the policy. The hypothesis such a study tests is whether the separation can tease apart what an agent's actions in a particular state accomplish and what the agent needs to accomplish to get to a goal from a given state. This hypothesis is evaluated by conducting experiments on the Fetch and Hand domains and evaluating how quickly the goal-conditioned policy is learned and whether it generalizes to goals in parts of the state space that it was not trained on.

**Summary Of The Review:**

The paper presents a valid hypothesis: that separating what the effect of an action will be in a given state and what the agent needs to do to solve a task should be separated while learning in order to aid the speed of learning as well as generalization. To do so the paper presents the separation of the value network into two networks whose vector predictions are combined to predict the value.
The downstream effects of this new architecture are validated experimentally fairly well. However, whether the new architecture actually learns the hypothesized separation is not evaluated. Additionally, the writing can be tightened up a bit.

---

> ### Author Response · Authors · 2021-11-23
> **response (1/3)**
>
> We thank the reviewer for the feedbacks and the comments and address the rest of questions below.
>
> ## New: Added A 2D Illustrative Example
>
> > Specifically, seeing an experiments that learns a vector field like Fig 2 (b) in actuality would have strengthened the case for the paper quite a bit.
> > As it stands, this paper shows some interesting empirical results with a new network architecture and presents a credible hypothesis for why this architecture might be helping learning, but does not dig deep enough to validate the hypothesis.
>
> We have included an experiment in a 2D maze in Section 4.5 in the updated manuscript. The state space is $\mathbb{R}^2$, which allows us to directly visualze both $\vec f$ and $\vec \varphi$. Our results show that the length of the learned $\vert \vec\varphi(s,g) \vert$ (black arrow, Fig 10b) roughly corresponds to the distance between $s$ and $g$.
>
> We further compare (same figure) the the angle $\angle_{f, \varphi}$ between that of the optimal action (from the policy) $\vec f(s, a^*)$ and a sub-optimal, random action $\vec f(s, a_\text{rand})$. We show that on this simple domain, the $\vec f(s, a^*)$ better aligns with $\vec \phi(s, g)$. Note that there is an offset of 90 degrees in the relative angle between $\vec f$ and $\vec \varphi$, that exists. This is because in this domain, the maximum value the agent can get is not 1, but zero. In generally, $\vec f$ for the optimal action will not be perfectly aligned with $\vec \phi$ -- this offset is a redundant degree of freedom.
>
> ## We Keep The Policy Network The Same
>
> > The paper mentions that the value network is decomposed into the bilinear networks the paper studies. But how are the policy networks handled? Are they keeping their monolithic structure? If so, why? These questions are not answered in the paper effectively.
>
> We only decompose the value network. The policy network is kept monolithic the same as the baseline. Methodologically because DDPG is an off-policy learning algorithm, we want to see if more generalizable value network can produce a more generalizable policy. For this reason we focus our experiment only on changing the value network and keep the policy the same.
>
> The value network outputs a single scaler, which is why we can decompose it into a bilinear map (to a scaler). The action is a vector however, making it unclear how to make it work with a scheme such as $\vec a = f(s)^T \varphi(g) (?)$.
>
> ## Summary of Benchmark Environments
>
> > It also does not attempt to generalize and extract insights beyond what are applicable to the few domains that it evaluates on.
>
> We appreciate this comment. To put things into perspective, we do include a fairly extensive set of testing environments. In fact, we test **more standardized multi-goal domains** than a number of recent publications. In total, we test on **4 Fetch robot manipulation tasks**, **8 Shadow hand tasks**, and 4 more from an extended fetch robot manipulation suite.
>
> We summarized how many domains some of the papers include in the table bellow:
>
> | Reference                           |               Domains Included |
> | ----------------------------------- | -----------------------------: |
> | Hindsight Experience Replay [1]     |                      Fetch x 4 |
> | Plappert, M. et al. (2018) [2]      |    Fetch x 4, Shadown Hand x 8 |
> | MEGA (2020) [3]                     |           Fetch x 2 + Maze x 2 |
> | Mapping State-space (2019) [4]      |                      Fetch x 4 |
> | Bi-linear Value Networks (Ours)     |   **Fetch x 4, Shadown Hand x 8, Fetch-Extension x 4** |
>
> [1] Andrychowicz, M. et al. (2017) ‘Hindsight Experience Replay’, NeurIPS 2017, Available at: http://arxiv.org/abs/1707.01495.
>
> [2] Plappert, M. et al. (2018) ‘Multi-goal reinforcement learning: Challenging robotics environments and request for research’, arXiv preprint arXiv. Available at: https://arxiv.org/abs/1802.09464.
>
> [3] Pitis, S. et al. (2020) ‘Maximum Entropy Gain Exploration for Long Horizon Multi-goal Reinforcement Learning’, ICML 2020, Available at: http://arxiv.org/abs/2007.02832.
>
> [4] Huang, Z., Liu, F. and Su, H. (2019) ‘Mapping State Space using Landmarks for Universal Goal Reaching’, NeurIPS 2019. Available at: http://arxiv.org/abs/1908.05451.

---

> ### Author Response · Authors · 2021-11-23
> **response (2/3)**
>
> ## Updated Writing In New Draft
>
> > The problem setup seems imprecise. A reader not familiar with reinforcement learning might not be able to parse this section. Specifically the structure of the reward function is not clarified, not to mention the range of discounting is unclear.
>
> We appreciate this feedback, and have updated the the preliminary section and the experimental setup section to include:
> 1. A new subsection on Multi-goal Reinforcement learning, where we talk about common reward structure and the discount factor mentioned above.
> 2. A new section in the appendix, that covers the detailed environment specification.
>
> **We reproduce the details here:** The reward $R(s,a,g) = 0$ when the agent reaches the goal, otherwise $R(s,a,g) = -1$. This reward function is canonical in multi-goal RL benchmark [5]. We follow [5] to set the discount factor as 0.98.
>
> [5] Plappert, M. et al. (2018) ‘Multi-goal reinforcement learning: Challenging robotics environments and request for research’, arXiv preprint arXiv. Available at: https://arxiv.org/abs/1802.09464.
>
> ## Clarifying Prior Works and Parameterization
>
> > In the experimental section, it refers to [2] as UVFA and [1] as a bilinear network, while the title of paper [1] is "Universal Value Function Approximators". It does so in other parts of the paper as well, where the implication becomes that a UVFA has to be a monolithic network.
>
> We appologize for the confusion, and have updated the text with the following convention:
> 1. A monolithic universal value function approximator, now referred to as `UVFA (monolithic)`
> 2. A Bi-linear decomposition proposed in the same UVFA paper, now referred to as `UVFA (bilinear)`. This formulation takes the form of $Q = f(s, a)^\top \cdot \varphi(g)$.
> 3. Our method, `Bilinear Value Networks (Ours)` or `BVN (Ours)`, that is $Q = f(s, a)^\top \cdot \varphi(s, g)$.
>
> The authors of HER [2] employs the monolithic value function from UVFA.
>
> We summarize the related works and parameterization in the following table:
>
> | Reference                                                     |               Parameterization |
> | ------------------------------------------------------------- | -----------------------------: |
> | Monolithic UVFA [1, 2]                                        |                     $f(s,a,g)$ |
> | Mirowski et. al. [3]                                          |        $\mathrm{RNN}(s, a, g)$ |
> | Bilinear UVFA (lowrank) [2] and UVSF (successor features) [4] |           $f(s, a) \cdot w(g)$ |
> | Proto-value function [5]                                      | $V(s) = \sum \lambda_i V^i(s)$ |
> | State-predictive representations [6]                          |   $Q(s,a) = \sum c^i Q^i(s,a)$ |
> | Bilinear value network (ours)                                 |  $f(s, a) \cdot \varphi(s, g)$ |
>
> Bilinear UVFA (the low-rank parameterization) is coverd in Schaul et al 2015. UVSF additionally imposes a training objective that enforces the successor feature interpretation.
>
> Our architecture is more sample efficient than monolithic UVFA,and prior decompositions that have been proposed.
>
> [1] Andrychowicz, Marcin, et al. "Hindsight experience replay." NeurIPS (2017).
>
> [2] Schaul, Tom, et al. "Universal value function approximators." *International conference on machine learning*. PMLR, 2015.
>
> [3] Mirowski, Piotr, et al. "Learning to navigate in cities without a map." *Advances in Neural Information Processing Systems* 31 (2018): 2419-2430.
>
> [4] Borsa, Diana, et al. "Universal successor features approximators." ICLR (2019).
>
> [5] Mahadevan, Sridhar. "Proto-value functions: Developmental reinforcement learning." *Proceedings of the 22nd international conference on Machine learning*. 2005.
>
> [6] Singh, Satinder P., et al. "Learning predictive state representations." *Proceedings of the 20th International Conference on Machine Learning (ICML-03)*. 2003.

---

> ### Author Response · Authors · 2021-11-23
> **response (3/3)**
>
> ## Updating Discussion
>
> > The discussion section seems like an extension of the experimental section. There is a variety of possible generalizations and implications of this work that should be discussed which are ignored in the discussion section for going over a few more experimental details.
>
> We have moved the original discussion section to the appendix, and rewrote the discussion section to think beyond what is covered in the submission. One way is to leverage the decomposition to facilitate transferring across different robots' morphology and goal spaces. Another way is to look for ways to pre-train a shared $f$ for the same robot, and learn different $\varphi$ for different tasks (i.e., domain goal spaces).
>
> ## MISC
>
> > The attribution of [2] should be for the NeurIPS version, rather than the ArXiv one.
>
> We have updated the reference formatting.
>
> > The idea of separating what can be done in a state and what needs to be done for a task bears some resemblance to the Q(s, s') learning in [3]. Perhaps it might be useful for discussion.
>
> The main difference of our method w.r.t to Edwards et al (2020) is that we do not change the formulation of the Q network, and do not need a forward and an inverse dynamics model to estimate the value.

---

### Official Review · Reviewer_vt3X · 2021-11-02

**Correctness:** 4
**Technical Novelty And Significance:** 2
**Empirical Novelty And Significance:** 3
**Recommendation:** 6
**Confidence:** 3

**Main Review:**


The paper proposes a modification to UVFAs that split them into two networks, each responsible for learning a particular "concept". On the one hand, the approach is a slight modification to the kind of architecture proposed in Schaul et al (2015), which the paper acknowledges. Having said that, the interpretation of the architecture is certainly interesting and, to the best of my knowledge, novel. Furthermore, in this instance, I feel like the simplicity and small change required is actually a strength of the paper, since the approach will be easy to integrate into existing codebases, which will help with adoption.

One particular thought I had when reading the paper that continues to nag at me is the relative contribution of the two vectors. Intuitively, it feels like $\phi$ is much more important than $f$. For example, in Figure 2, $f$ directs the agent passed the obstacle. While it makes sense in general not to collide with an obstacle, why should that be a preferred direction. The goal could be to the left of the agent instead, which would mean it should actually be moving away from the obstacle, and not around it. This would obviously require $\phi$ to do all the "heavy lifting" but I can't help wondering when $\phi$ isn't the vector that is most important. As a simple example, we could imagine an unbounded 2D plane with some goal. Then the direction proposed by $\phi$ is the only thing that actually matters and $f$ can be completely ignored. I may be missing something, but I feel (at least in the navigation case which is easier to reason about) that this would generally be the case.  It would be great to have some kind of experiment (even in a toy domain) that speaks to this. I'm imagining something where an agent is trained and then the average magnitude (or something like that) of the vectors is recorded to try measure the influence of each component.

Along similar lines, it would also be interesting to see a setup (again, this could be a synthetic environment) where $\phi$ directs the agent towards the goal, but that's actually the wrong thing to do (because there is a dead-end or pit of lava or something to that effect). I would be curious to see the behaviour elicited in this case.

While the experiments were generally well-executed, I feel like they could have benefited from at least one more baseline. While the comparison to UVFAs is natural, it would have been nice to see how it performed against more recent approaches like SAC (Levy et al, 2019) or HIRO (Nachum et al, 2019)


Though it is not directly related, the approach of dividing into the two concepts of "where to go" and "what to do" put me in mind of a number of different goal-oriented approaches and it may be worth including a discussion of some in the related work.  For example, the separation of concerns is reminiscent of Edwards et al (2020) who learn a state-next state value function and then subsequently learn how to act to reach that. A slightly different take is to have a value function propose a subgoal region to visit, and then a controller that attempts to achieve that (Levy et al, 2019; Nachum et al, 2018; Hejna et al, 2020; Christen et al, 2021)

Minor comments:

  1. It is very difficult to see the vector field in Figures 1 and 2 because the arrows are a very light grey
  2. Figure 3 appears before Figure 2
  3. \citep is used in the first sentence of Section 2.1 instead of \citet
  4. Last sentence on page 5: "Euclidean"

References

1. Levy, Andrew, et al. "Learning Multi-Level Hierarchies with Hindsight." International Conference on Learning Representations. 2019.
2. Nachum, Ofir, et al. "Near-Optimal Representation Learning for Hierarchical Reinforcement Learning." International Conference on Learning Representations. 2019
3. Edwards, Ashley, et al. "Estimating q (s, s’) with deep deterministic dynamics gradients." International Conference on Machine Learning. PMLR, 2020.
4. Hejna, Donald, Lerrel Pinto, and Pieter Abbeel. "Hierarchically decoupled imitation for morphological transfer." International Conference on Machine Learning. PMLR, 2020.
5. Christen, Sammy, et al. "Learning functionally decomposed hierarchies for continuous control tasks with path planning." IEEE Robotics and Automation Letters 6.2 (2021): 3623-3630.


**Summary Of The Paper:**

The paper proposes a modification to goal-oriented universal value functions that split the neural network into two parts. One network accepts states and actions and outputs a vector representation, while the other outputs a vector given a state and goal. Part of the contribution of the paper is the interpretation of these two vectors --- the first captures the notion of what actions can be taken at a given state, while the second represents the direction the agent should move towards to achieve the goal from its current state.  Empirical results on several simulated robotic domains demonstrate that the method outperforms other UVFA architectures, while ablations are provided to capture the most important aspects of the method.

**Summary Of The Review:**

Though the actual method proposed is fairly incremental, the interpretation of it is novel and its simplicity means it could likely be incorporated into the goal-oriented learning toolkit quite easily. The experiments were all well executed and showed improved empirical performance, and ablation studies are provided to help understand exactly which design decisions are responsible for improved performance.

***********************
POST REBUTTAL
***********************

The newly-added experiment in 4.5 strengthens the paper in my view, since it highlights the particularities of the method. I think the approach is an interesting one and could be valuable to the existing body of work on goal-oriented learning. It is also conceptually simple enough that it could easily be widely adopted and used as baselines or built upon going forward

---

> ### Author Response · Authors · 2021-11-23
> **response (1/3)**
>
> We greatly appreciate Reviewer vt3X's review and address the concenrs below:
>
> ## Additional Baselines To Show Generalizability
>
> We agree with your assessment:
>
> > I feel like they could have benefited from at least one more baseline. While the comparison to UVFAs is natural, it would have been nice to see how it performed against more recent approaches like SAC (Levy et al, 2019)
>
> In the updated draft that we have included two more baselines: Soft-actor critic [1], and twin-delayed deterministic deep policy gradient (TD3) [2] in Section B. Our results showed  that our bilinear value network outperforms all the baselines. We found that DDPG+HER performs better than TD3+HER and SAC+HER, and Bilinear DDPG+HER (our method) is superior to DDPG+HER.
>
> [1] Haarnoja, Tuomas, et al. "Soft actor-critic: Off-policy maximum entropy deep reinforcement learning with a stochastic actor." International conference on machine learning. PMLR, 2018.
> [2] Fujimoto, Scott, Herke Hoof, and David Meger. "Addressing function approximation error in actor-critic methods." International Conference on Machine Learning. PMLR, 2018.
>
> Discussion of HAC (Levy et al 2019): applies multi-goal actor-critic methods with monolithic value functions as a messaging passing interface in hierachical RL, where the high-level policy's outputs are the inputs (i.e., goals) of the low-level policy. This work is orthogonal to ours since our bilinear value network can be a drop-in replacement of their monolithic value function.
>
> Difference from Edwards et al (2020): in that we separate local and global components in the Q-function's architecture instead of its formulation. We do not need a forward and an inverse dynamics model to estimate the optimal next state and the corresponding action.
>
> That being said, we agree that in the abstract sense Levy 2019 and Edwards et al 2020 share the same spirit with our method and we will include them into our related work section.
>
> ## Relative Contributions/Interpretation of $f(s, a)$ and $\varphi(s, g)$
>
> **Relative contribution:** These two functions, $f(s, a)$ and $\varphi(s, g)$ interact in a non-trivial way to approximate the value function. The pointing of the vector $\vec f(s, a)$ depends on the particular action $a$. $\max_a Q(s, a, g)$ is maximized when $\vec f$ **aligns perfectly with the field $\vec \varphi(s, g)$**. Solely using $\varphi(s,g)$ cannot model the actions' contribution to the Q-value,
>
> **Interpretation:** To answert the reviewer's question on the interpretation on $f$ and $\varphi$ , we would like to introduce some important properties of these two functions.
>
> It is important to keep in mind the exact parameterization of these two functions is **unidentifiable** in the sense that one can insert an arbitrary non-singular real vector $\vec\lambda$ to $\lambda f$ and $\lambda -1 \varphi$, and it would not change the value estimation. $\lambda$ corresponds to an unconstrained, free degrees of freedom in the rotation of the field.
>
> This is similar to what is mentioned in the Duelling Network paper by Wang *et al*. A key implication, is that the **exact direction** of the vector $\vec f(s, a)$ can be different random-seed-by-random-seed, as long as it matches the corresponding $\vec \varphi(s, g)$. We also note that we do not impose any constraint during learning to enforce a particular interpretation.
>
> Now, we answer the specfiic reviewer's questions below
>
> > For example, in Figure 2, f directs the agent passed the obstacle. While it makes sense in general not to collide with an obstacle, why should that be a preferred direction.
>
> This is because that the optimal actions' $f(s,a)$ embeddings should align with $\varphi(s, g)$ so the $Q(s, a, g)$ is maximized. The direction to pass the obstacle will lead to higher Q-value than that to hit the wall.
>
> > Along similar lines, it would also be interesting to see a setup (again, this could be a synthetic environment) where ϕ directs the agent towards the goal, but that's actually the wrong thing to do (because there is a dead-end or pit of lava or something to that effect). I would be curious to see the behaviour elicited in this case.
>
> If the action $a$  that results in maximum $f(s,a)^T \varphi(s,g)$ directs the agent to the wall, this implies that the learned  $Q(s,a,g)$ might be wrong and need more training data to estimate the Q-value accurately.

---

> ### Author Response · Authors · 2021-11-23
> **response (2/2)**
>
> ## A New, Illustrative Toy Domain
>
> We have included an experiment in a 2D maze in Section 4.5 in the updated manuscript. The state space is $\mathbb{R}^2$, which allows us to directly visualze both $\vec f$ and $\vec \varphi$. Our results showed that the length of the learned $\vert \vec\varphi(s,g) \vert$ (black arrow, Fig 12b) roughly corresponds to the distance between $s$ and $g$.
>
> We further compare (same figure) the the angle $\angle_{f, \varphi}$ between that of the optimal action (from the policy) $\vec f(s, a^*)$ and a sub-optimal, random action $\vec f(s, a_\text{rand})$. We show that on this simple domain, the $\vec f(s, a^*)$ better aligns with $\vec \phi(s, g)$. Note that there is an offset of 90 degrees in the relative angle between $\vec f$ and $\vec \varphi$, that exists. This is because in this domain, the maximum value the agent can get is not 1, but zero. In generally, $\vec f$ for the optimal action will not be perfectly aligned with $\vec \phi$ -- this offset is a redundant degree of freedom.
>
> --------
>
> The authors hope that these comments address some of reviewer vt3X's concerns. We have addressed the reviewer's minor comments in the updated manuscript. Also note that we have submitted our experimental code base as the supplement, which we intend to opensource upon acceptance.

---

### Official Review · Reviewer_ZNRs · 2021-11-03

**Correctness:** 3
**Technical Novelty And Significance:** 3
**Empirical Novelty And Significance:** 3
**Recommendation:** 6
**Confidence:** 4

**Main Review:**

The main contribution of the paper is proposed modification of the universal value function approximation implementation by making the goal-dependent component also depend on the state so that now it is f(s,a)*\phi(s,g) instead of f(s,a)*\phi(g). The improvement is simple and well motivated. However, the method is not explained very well. After the idea is presented in section 3, nothing is said about how the method is implemented (how the functions are parameterized and trained?). Are f and \phi the networks? What kind of networks? How states, actions and goals are processed in them? As a consequence, it is also not clear to me how the comparison to the baselines is done. For example, is the number of training parameters the same? Besides, it would be informative to include the information about the state, action and goal representations and their dimensions as it would help to understand the experiments like the ones in Figure 9.

My other concern is the novelty of the proposed method. It seems to me that the decompositions into a global component that depends on goal and state and local that looks at state and action has been already used in the literature (maybe without stating it as the main contribution). For example, see such decomposition is use for transfer between navigation tasks as in the network structure of “Learning to Navigate in Cities Without a Map” by Mirowski et al. 2019 (figure 2b). Could the authors comment on the similarities and differences between these methods? How would the proposed method perform compared to such a baseline?

The main experiments clearly demonstrate the usefulness of the method for the purpose of data-efficiency and transfer to unseen goals. The analysis of the method is useful, but does not provide sufficient light into what kind of decompositions are learnt. Does the training always result in similar decomposition? How can it be illustrated that the decomposition has similar meaning as said in the motivations (figure 2)? One of the ablation studies that is supposed to answer the question “Does the performance gain result from a larger model?” does not answer that question in my opinion. That experiment shows that the particular architecture modification -- concatenation of the outputs of f and \phi followed by an MLP -- does not further improve the performance. To answer the original question the authors should: 1) provide information about the number of training parameters in the baseline methods and the proposed method, 2) try to increase the number of parameters without architecture modifications (for example, by increasing the number of hidden units in the existing layers). The ablations with the decomposition f(s,a,g)*\phi(g) seems to me not very informative because such a baseline is quite logically can’t improve the performance as it contains the full UVFA f(s,a,g) which is then multiplied by the value depending on the goal which does not have any particular motivation. Also, maybe other decompositions could be considered, such as f(s,a)*\phi(a,g)?

Pros:

- The proposed idea is simple and elegant.

- The environments and the design of the experiments chosen for a study seem to be interesting and challenging.

- The benefits of the method in terms of data efficiency and transfer to new goals are clear.

Cons:

- Details about the implementation of the method and comparison to the baselines are missing.

- Novelty of the method might be limited given that some prior work already looked at decomposing learning of the local component (depending on state and action) and global component (depending on state and goal).

- The analysis of the method is not convincing. Moreover, one of the ablation studies seems to me to be incorrectly designed and another one not very informative.

Other comments:

- Some phrases are unclear or not well written, for example “transfer to new goals type of transfer to new goals”, “generalizability improves sample efficiency”.

- The references of the paper need formatting.


**Summary Of The Paper:**

This paper proposes a new decomposition for the universal value function that disentangles local and global components. The global component depends on the goal and state and tells the agent “where to go” and the local component depends on the state and action and tells the agent “what to do”. The authors conduct a set of experiments on a number of environments to demonstrate that this decomposition results in more data efficiency and helps to generalise to the new goals.

**Summary Of The Review:**

The proposed method is simple and elegant, but it is unclear to me how it relates to some of the previous literature that proposed similar decomposition through the network structure. Some details of the method implementation and experiments are missing. The main experiments are convincing and well designed, but the analysis is not sufficient and does not provide enough insight into the behaviour of the method.

---

> ### Author Response · Authors · 2021-11-23
> **response (1/3)**
>
> Thank you for your thorough review. We find your comments helpful, and have uploaded an updated draft that addresses the following issues:
>
> - provide implementation details of our method and the baselines and the details of the experimental setup
> - provide information about the number of training parameters in the baseline methods and the proposed method
> - match the parameters without architecture modifications
> - discuss Mirowski et al. 2019
> - add the experiments of decompositions: $f(s,a)*\varphi(a,g)$
> - added the specification of all test domains in Appendix 1.
>
> ## On Novelty
>
> > My other concern is the novelty of the proposed method. It seems to me that the decompositions into a global component that depends on goal and state and local that looks at state and action has been already used in the literature (maybe without stating it as the main contribution). For example, see such decomposition is use for transfer between navigation tasks as in the network structure of “Learning to Navigate in Cities Without a Map” by Mirowski et al. 2019 (figure 2b). Could the authors comment on the similarities and differences between these methods? How would the proposed method perform compared to such a baseline?
>
> The architecture proposed in Mirowski et al can be represented as $RNN(env_i(s,g), conv(s), a)$, where $env_i$ and $conv$ are neural networks. For each city $i$, they process state ($s$) and goals ($g$) by a different $env_i$, and concatenate the output $env_i(s,g)$, $conv(s)$, and $a$ as the inputs of an RNN. Thus, their architecture is more akin to a monolithic value function (i.e., $Q(s,a,g) = f(s,a,g)$) in the sense that they feed the preprocessed inputs (i.e., $env_i(s,g)$, $conv(s)$) to a fully-connected RNN module. In contrast, instead of feeding the concatenated preprocessed input, our main contribution is to show that decomposing value outputs to the dot-product of two vector-valued functions $Q(s,a,g) = f(s,a)·\varphi(s,g)$ is largely better.
>
> Our bilinear decomposition is novel in relation to prior works that also linearly decompose the value function because all three, (i) state-predictive representation, (ii) successor features, and (iii) proto-value functions, decompose the value function into (s, a)·(g). In our experiment we show that these naive decomposition schemes cause the performance to drop on the Fetch manipulation domains in comparison to the baseline that uses a monolithic value network (one that does Q(s, a, g)).
>
> These works tested their algorithms on simple, 2D navigation tasks, where the state and goal space live on a low-dimensional 2D manifold. **Structurally, the robotic manipulation tasks** we use are **more complex than 2D navigation**. In pick-and-place, for example, the agent needs to touch the object in order for the object to move, whereas in reach, there are no such issues. The state manifold is also a much higher dimension. The ShadowHand domains (we test 8 of them in the paper) are `(61,)` dimensions. The Fetch domains are `(25,)` (Reach is `(15,)`). This added dimensionality is in addition to the added complexity due to physical contact.
>
> We summarize the related works and parameterization in the following table:
>
> | Reference                                           |               Parameterization |
> | --------------------------------------------------- | -----------------------------: |
> | Monolithic UVFA [1, 2]                              |                     $f(s,a,g)$ |
> | Mirowski et. al. [3]                                |        $\mathrm{RNN}(s, a, g)$ |
> | Bilinear UVFA [2] and UVSF (successor features) [4] |           $f(s, a) \cdot w(g)$ |
> | Proto-value function [5]                            | $V(s) = \sum \lambda_i V^i(s)$ |
> | State-predictive representations [6]                |   $Q(s,a) = \sum c^i Q^i(s,a)$ |
> | Bilinear value network (ours)                       |  $f(s, a) \cdot \varphi(s, g)$ |
>
> Note that Bilinear UVFA uses the same parameterization as that of UVSF, but UVSF additionally imposes a training objective that enforces the successor feature interpretation. Our architecture is more sample efficient than monolithic UVFA, and prior bilinear decompositions that have been proposed.
>
> [1] Andrychowicz, Marcin, et al. "Hindsight experience replay." NeurIPS (2017).
>
> [2] Schaul, Tom, et al. "Universal value function approximators." *International conference on machine learning*. PMLR, 2015.
>
> [3] Mirowski, Piotr, et al. "Learning to navigate in cities without a map." *Advances in Neural Information Processing Systems* 31 (2018):
>
> [4] Borsa, Diana, et al. "Universal successor features approximators." ICLR (2019).
>
> [5] Mahadevan, Sridhar. "Proto-value functions: Developmental reinforcement learning." *Proceedings of the 22nd international conference on Machine learning*.
>
> [6] Singh, Satinder P., et al. "Learning predictive state representations." *Proceedings of the 20th International Conference on Machine Learning (ICML-03)*.

---

> > ### Comment · Reviewer_ZNRs · 2021-11-25
> > **Thank you for the clarifications and updating the score**
> >
> > I would like to thank the authors for their detailed answers. In particular, the point about the novelty and related work is more clear to me. I looked at [3] again and I agree with the interpretation, although I feel f(g(s,g),s,a) would be a bit more precise. I think the discussion of various design decisions like this in the related work would be informative for new readers.
> >
> > Thanks for providing the additional details, I encourage you to keep them for the final manuscript. In particular, I appreciate the addition of Figure 10 that helps to understand the method.
> >
> > In the view of the improvement and clarifications, I feel that I can increase my score. The proposed method is simple and intuitive and it could be a nice addition to the conference.

---

> ### Author Response · Authors · 2021-11-23
> **response (2/3)**
>
>
> ## Clarity And Implementation Details
>
> > However, the method is not explained very well. After the idea is presented in section 3, nothing is said about how the method is implemented (how the functions are parameterized and trained?). Are f and \phi the networks? What kind of networks? How states, actions and goals are processed in them? As a consequence, it is also not clear to me how the comparison to the baselines is done. For example, is the number of training parameters the same? Besides, it would be informative to include the information about the state, action and goal representations and their dimensions as it would help to understand the experiments like the ones in Figure 9.
>
> **We have updated the manuscript, and reproduce some of the details below:**
>
> - **Model architecture**: Both $f$ and $\phi$ are three-layer networks with 256 neurons. The current experiment represents the Q-function as a three-layer network with 256 neurons in the baselines. **To match the total number of parameters, we updated our manuscript with experiments that use an MLP with 176 latent neurons for $f$ and $\phi$ in our method**.
>
> - **Training approach**: Algorithm-wise we use the standard DDPG + HER implementation adapted from here: [https://github.com/TianhongDai/hindsight-experience-replay]. This is a state-of-the-art implementation that is used as a base algorithm in recent publications, such as [7, 8].
>
> - **State Representation**: We have included detailed specs of the domains in Appendix 1, which we reproduce below:
> 	- FetchReach: $s \in \mathbb{R}^{10}$, $a \in \mathbb{R}^3$, $g \in \mathbb{R}^3$
> 	- FetchPush: $s \in \mathbb{R}^{25}$, $a \in \mathbb{R}^3$, $g \in \mathbb{R}^3$
> 	- FetchPickAndPlace: $s \in \mathbb{R}^{25}$, $a \in \mathbb{R}^4$, $g \in \mathbb{R}^3$
> 	- FetchSlide: $s \in \mathbb{R}^{25}$, $a \in \mathbb{R}^3$, $g \in \mathbb{R}^3$
> 	- DrawerClose: $s \in \mathbb{R}^{25}$, $a \in \mathbb{R}^3$, $g \in \mathbb{R}^3$
> 	- DrawerOpen: $s \in \mathbb{R}^{25}$, $a \in \mathbb{R}^3$, $g \in \mathbb{R}^3$
> 	- ShadowHandBlockRotateZ: $s \in \mathbb{R}^{61}$, $a \in \mathbb{R}^{20}$, $g \in \mathbb{R}^{7}$
> 	- ShadowHandBlockRotateParallel: $s \in \mathbb{R}^{61}$, $a \in \mathbb{R}^{20}$, $g \in \mathbb{R}^{7}$
> 	- ShadowHandBlockRotateXYZ: $s \in \mathbb{R}^{61}$, $a \in \mathbb{R}^{20}$, $g \in \mathbb{R}^{7}$
> 	- ShadowHandBlockRotate: $s \in \mathbb{R}^{61}$, $a \in \mathbb{R}^{20}$, $g \in \mathbb{R}^{7}$
> 	- ShadowHandEggRotate: $s \in \mathbb{R}^{61}$, $a \in \mathbb{R}^{20}$, $g \in \mathbb{R}^{7}$
> 	- ShadowHandEgg: $s \in \mathbb{R}^{61}$, , $a \in \mathbb{R}^{20}$, $g \in \mathbb{R}^{7}$
> 	- ShadowHandPen: $s \in \mathbb{R}^{61}$, , $a \in \mathbb{R}^{20}$, $g \in \mathbb{R}^{7}$
> 	- ShadowHandPenRotate: $s \in \mathbb{R}^{61}$, , $a \in \mathbb{R}^{20}$, $g \in \mathbb{R}^{7}$
>
> [7] Pitis, Silviu, et al. "Maximum entropy gain exploration for long horizon multi-goal reinforcement learning." *International Conference on Machine Learning*. PMLR, 2020.
>
> [8] Zhang, Lunjun, Ge Yang, and Bradly C. Stadie. "World model as a graph: Learning latent landmarks for planning." *International Conference on Machine Learning*. PMLR, 2021.
>
> - **State, action, and goal**: States ($s$), actions ($a$), and goals ($g$) are all vectors with different dimensions. For the baseline $Q(s,a,g)$, we follow the practice in typical multi-goal RL implementation [9], concatenating $s$, $a$, and $g$ and then feeding to the network. In our method, wee concatenate $(s, a)$ and $(s, g)$ and feed to $f$ and $\phi$ respectively.
>
> [9] Plappert, Matthias, et al. "Multi-goal reinforcement learning: Challenging robotics environments and request for research." arXiv preprint arXiv:1802.09464 (2018).
>
> We have also added the full table for all of the environments in Section A.1 of the appendix. To make it easy to reproduce our results, we have uploaded our experiment code base to the supplementary material. We plan to release this code as soon as our paper is accepted.

---

> ### Author Response · Authors · 2021-11-23
> **response (3/3)**
>
>
> ## Toy Domain and Illustrative Example
>
> We have included an experiment in a 2D maze in Section 4.5 in the updated manuscript. The state space is $\mathbb{R}^2$, which allows us to directly visualze both $\vec f$ and $\vec \varphi$. Our results showed that the length of the learned $\vert \vec\varphi(s,g) \vert$ (black arrow, Fig 12b) roughly corresponds to the distance between $s$ and $g$.
>
> We further compare (same figure) the the angle $\angle_{f, \varphi}$ between that of the optimal action (from the policy) $\vec f(s, a^*)$ and a sub-optimal, random action $\vec f(s, a_\text{rand})$. We show that on this simple domain, the $\vec f(s, a^*)$ better aligns with $\vec \phi(s, g)$. Note that there is an offset of 90 degrees in the relative angle between $\vec f$ and $\vec \varphi$, that exists. This is because in this domain, the maximum value the agent can get is not 1, but zero. In generally, $\vec f$ for the optimal action will not be perfectly aligned with $\vec \phi$ -- this offset is a redundant degree of freedom.
>
>
> ## Additional Ablation Studies
>
> To rule out the possibility that the performance gain comes from larger models, we have updated the manuscript with new results using smaller models in our method. In the updated experiments (Sections 4.2, 4.3, and 4.4), $f$ and $\phi$ are both 3-layer neural network with 176 neurons so the total number of parameters match the baselines' models using 256 latent neurons.
>
> The results show that bilinear value network with less parameters matches the reported performance, and still outperforms the baselines.
>
> ## Open-source Release
>
> We have uploaded our experimental code base as supplementary, and plan to release the code as soon as the paper is accepted.
>
>
> ## Other comments
>
> We have updated our manuscript according to your suggestions.

---

### Official Review · Reviewer_AnrU · 2021-11-06

**Correctness:** 3
**Technical Novelty And Significance:** 2
**Empirical Novelty And Significance:** 2
**Recommendation:** 6
**Confidence:** 3

**Main Review:**


PRO

* well written and easy to follow
* simple method with good motivation
* good ablation studies and discussion for the different aspects of the proposed method

CONS
* it's not discussed why the method performs worse or on par in some cases
* only tests the method with DDPG and in two domains. So it's unclear how general the proposal is

There are a number of open questions

I would have hoped to see a more general evaluation though as essentially this seems to be a very general method that could plugin to value function approaches more generally but also into different actor-critic methods. Why did you choose to focus only on two domains and DDPG? Does the approach generalize to other domains and other actor-critic MGRL methods?

Is it important that this is bilinear?

What are the NN dimensions of the value networks? Does a change here have any effect?

Is adding more structure like an MLP really fair? Clearly you add a lot of structure through the bi-linear constraint. So just making it more complex and adding an MLP doesn't really address the question of whether the larger model is giving the performance gain

**Summary Of The Paper:**

The paper proposes a new approach to Universal Value Functions in Reinforcement Learning by decomposing monolithic universal value functions into two components and exploring different ways of combining the results of the decomposition. The paper shows that this leads to better sample efficiency and has desirable properties for generalization.

**Summary Of The Review:**

The paper is well written and does tackle an interesting and important part of Reinforcement Learning.
The idea presented is quite simple but does show promising results in the experiments.

Authors do a good job of ablating and discussing different choices of the systems

Some information seems missing around generality of the method.

---

> ### Author Response · Authors · 2021-11-23
> **response (1/2)**
>
> The authors appreciate reviewer AnrU's helpful feedback! We have updated the draft and will produce some of the updates below:
>
> ## On The Generalizability of The Method
>
> > only tests the method with DDPG and in two domains. So it's unclear how general the proposal is
>
> To clarify: we test on **more standardized multi-goal domains** than a number of recent publications. We test on 4 Fetch robot manipulation tasks, 8 Shadow hand tasks, and 4 more from an extended robot manipulation suite.
>
> We summarized the domains that recent publicatoins have experimented with:
>
> | Reference                           |               Domains Included |
> | ----------------------------------- | -----------------------------: |
> | Hindsight Experience Replay [1]     |                      Fetch x 4 |
> | Plappert, M. et al. (2018) [2]      |    Fetch x 4, Shadown Hand x 8 |
> | MEGA (2020) [3]                     |           Fetch x 2 + Maze x 2 |
> | Mapping State-space (2019) [4]      |                      Fetch x 4 |
> | Bi-linear Value Networks (Ours)     |   **Fetch x 4, Shadown Hand x 8, Fetch-Extension x 4** |
>
> 1. Andrychowicz, M. et al. (2017) ‘Hindsight Experience Replay’, NeurIPS 2017, Available at: http://arxiv.org/abs/1707.01495.
> 2. Plappert, M. et al. (2018) ‘Multi-goal reinforcement learning: Challenging robotics environments and request for research’, arXiv preprint arXiv. Available at: https://arxiv.org/abs/1802.09464.
> 3. Pitis, S. et al. (2020) ‘Maximum Entropy Gain Exploration for Long Horizon Multi-goal Reinforcement Learning’, ICML 2020, Available at: http://arxiv.org/abs/2007.02832.
> 4. Huang, Z., Liu, F. and Su, H. (2019) ‘Mapping State Space using Landmarks for Universal Goal Reaching’, NeurIPS 2019. Available at: http://arxiv.org/abs/1908.05451.
>
> ### Extensive Actor-Critic Algorithms
>
> In the updated draft that we have included two more baselines: Soft-actor critic [1], and twin-delayed deterministic deep policy gradient (TD3) [2] in Section B of the appendix. Our results showed  that our bilinear value network outperforms all the baselines. We found that DDPG+HER performs better than TD3+HER and SAC+HER, and Bilinear DDPG+HER (our method) is superior to DDPG+HER. Our findings on the comparison between DDPG, TD3, and SAC in multi-goal RL are consistent with [1].
>
> [1] Haarnoja, Tuomas, et al. "Soft actor-critic: Off-policy maximum entropy deep reinforcement learning with a stochastic actor." International conference on machine learning. PMLR, 2018.
> [2] Fujimoto, Scott, Herke Hoof, and David Meger. "Addressing function approximation error in actor-critic methods." International Conference on Machine Learning. PMLR, 2018.
> [3] Pitis, Silviu, et al. "Maximum entropy gain exploration for long horizon multi-goal reinforcement learning." International Conference on Machine Learning. PMLR, 2020.
>
> ## Why is The Method On-par Or Worse In Some Cases?
>
> In `Drawer-close` where our method performs slightly worse, the goal space is a single point. A.k.a there is just a single goal. This lack of diverse goals might be a contribution factor to the performance degradation, as a single, monolithic value function is sufficient in learning the value.
>
> Our method is only on-par with the base vanilla MLP value function in domains where exploration is a problem. `Fetch-slide` for example (and `Pen-rotate` and `Pen`), are known to be limited primarily by poor exploration. We qauntify the difficulty of exploration by goal-hitting rate in 1000 rollouts with random actions. `Fetch-slide` has the lowest goal-hitting rate (i.e., $0.01$ out of $[0.11, 0.06, 0.04]$ in other tasks). To make sure that our comparison is fair, we employed the gaussian exploration strategy used in the base DDPG/HER baseline. Mitigating the exploration issue is beyond the scope of this work.

---

> ### Author Response · Authors · 2021-11-23
> **response (2/2)**
>
> ## Other Implementation Details and Insights
>
> > What are the NN dimensions of the value networks? Does a change here have any effect?
>
> We use three layers of 256 neurons for all the baselines' Q-function. For our method, we represent both $f$ and $\varphi$ functions as a neural network with three layers of 256 neurons.
>
> To test the influence of architectural changes, we have updated our manuscript with the experiments of our bilinear value network with a smaller model that matches the total number of parameters of the baseline models. The results show that our method still outperforms the baselines. The smaller architecture's peformance matches the larger architecture. The results are reported in Section 4.2 and Section 4.3.
>
> > Is adding more structure like an MLP really fair? Clearly you add a lot of structure through the bi-linear constraint.
>
> **We made sure that our network has the same number of parameters as the baseline**. We do not introduce more parameters in our implementation.
>
> We do introduce an inductive bias via our bilinear decomposition, but it is quite easy to implement (very few code change), and as we have shown through extensive evaluation on Fetch and Shadow hand, the gain is not domain-specific. We believe this a useful prior for learning Q-values, similar to how convolution is a useful prior for image analysis.
>
> Methodologically, we also consider bilinear decomposition a unification of prior approaches that also decompose Q-values. Based on this view and the experimental results, we hope to communicate that bilinear decomposition is a simple and scalable technique, that has produced significant performance improvement on a variety of domains.
>
> We would love hear your updated opinion, and any question that arises.

---

### Decision · Program_Chairs · 2022-01-20

**Decision:**

Accept (Poster)

**Comment:**

This paper proposes a new bilinear decomposition for universal value functions.  The bilinear network has one component dependent on state and goal and another component that depends on state and action.  The experiments with the DDPG algorithm in robot simulations show that the proposed architecture improves performance data efficiency and task transfer over several baseline algorithms, including improvements on earlier bilinear decompositions.

The reviews noted several aspects of the paper could be improved, and the author response addressed several of these concerns. Multiple reviewers appreciated the insights from the experiment added in section 4.5 on a simple grid environment, which enabled a direct interpretation of the vector fields used in the method.  Several aspects of the presentation were clarified based on the reviewers comments. Additional details were also provided on the problem specification and the solution methods. During the discussion, the reviewers agreed that the revised paper presented a useful addition to the literature.

Four knowledgeable reviewers indicate to accept the paper for its contribution of an effective network architecture for a goal-conditioned universal value function approximator.  The paper is therefore accepted.